# Revisited Surgical Anatomy of the Left Colonic Angle for Tailored Carcinologic Colectomy: A Review

**DOI:** 10.3390/jpm13081198

**Published:** 2023-07-28

**Authors:** Roukaya Belhadjamor, Gilles Manceau, Benjamin Menahem, Charles Sabbagh, Arnaud Alves

**Affiliations:** 1Department of Digestive Surgery, University Hospital of Caen, CS 30001, CEDEX 9, 14033 Caen, France; belhadjamor_roukaya@hotmail.fr; 2Department of Digestive and Oncology Surgery, Assistance Publique Hôpitaux de Paris, Georges-Pompidou European Hospital, 75908 Paris, France; gilles.manceau@aphp.fr; 3Calvados Digestive Cancer Registry “ANTICIPE” U1086 INSERM, Team Ligue Contre le Cancer, Centre François Baclesse, University of Caen Normandy, 14000 Caen, France; alves-a@chu-caen.fr; 4Pôle de Formation et de Recherche en Santé, 2 rue des Rochambelles, 14032 Caen, France; 5Department of Digestive Surgery, University Hospital of Amiens, CHU Amiens Picardie, Rond point du Pr Cabrol, 80054 Amiens, France; sabbagh.charles@chu-amiens.fr; 6UR7518SSPC (Simplification des Soins des Patients. Chirurgicaux Complexes), Université Picardie Jules Verne, 80000 Amiens, France

**Keywords:** left colonic angle, anatomy, colonic cancer

## Abstract

Purpose: Although several types of surgical procedure have been advocated to date, the optimal resection of the left colonic angle in cancer treatment remains controversial. Located at the border of the transverse and descending colons, the anatomy of the left colonic angle is complex and characterized by numerous anatomic variations. Recent advances in preoperative (three-dimensional CT angiography with colonography) and/or intraoperative (indocyanine green staining) imaging have allowed for a better identification of these variations. Methods: We performed a methodological review of studies assessing the anatomical variations of the left colic artery. Results: While the left colonic angle is classically vascularized by branches of the superior and inferior mesenteric arteries, an accessory middle colonic artery has been identified from 6 % to 36% of cases, respectively, leading to their classification of five types. In the absence of a left colic artery, this artery becomes predominant. In parallel to the variations in the venous drainage of the left colonic angle, which has been classified into four types, new lymphatic drainage routes have also been identified via this accessory artery and the inferior mesenteric vein. Conclusions: Collectively, these newly obtained findings plead for preoperative identification in cases of cancer of the left colonic angle and a surgical strategy adapted to these anatomical variations.

## 1. Introduction

Left colonic angle cancer is rare, representing less than 8% of all colorectal cancers [1]. Compared to other colonic locations, left colonic angle cancer has a poorer prognosis due to (i) a high risk of occlusion, (ii) a more frequent metastatic stage at diagnosis, and (iii) a lower resectability [2,3,4]. While surgery for right and left colon cancers has been well established, there is currently no consensus regarding the type of elective colectomy to be performed for left colonic angle cancer [5,6,7]. In clinical practice, three types of colectomy can be discussed: left angular segmental colectomy, right colectomy extended to the left colonic angle (subtotal colectomy), and true left hemicolectomy. These alternatives are related to the anatomical features of the left colonic angle. Although it represents the anatomical part of the colon between the distal third of the transverse colon and the proximal third of the descending colon, the numerous variations in its vascularization and lymphatic drainage are not well known [8,9,10]. However, a better preoperative knowledge of these vascular anatomical variations could guide the surgeon in the type of surgical resection performed and decrease any intraoperative complications that may have arisen [11]. In light of recent surgical and radiological work, we proposed to review the surgical anatomy of the left colic angle to better understand the anatomical variations that are crucial for minimally invasive carcinologic colorectal surgery.

## 2. Methods

### 2.1. Study Selection

We conducted the systematic review in accordance with the PRISMA recommendations [12]. Articles included in the review were selected using the MEDLINE, Cochrane Central Register of Controlled Trials (CENTRAL), and Web of Science databases with the following keywords: left colic angle or splenic flexure colic angle and digestive surgery or colorectal surgery using the formula: (left colic angle OR splenic flexure colic angle) AND (digestive surgery OR colorectal surgery). The selection was restricted to articles indexed from database inception to 6 February 2023.

### 2.2. Inclusion and Exclusion Criteria

All human clinical trials and studies evaluating the anatomy of the left colonic angle were identified. The inclusion criteria were studies concerning adult patients who underwent colorectal surgery in the perspective of elective left colic abdominal surgery. No unpublished data nor data published only in the form of an abstract were used in the selection.

### 2.3. Data Reviewing and Extraction

Two investigators (RBA and BM) independently reviewed the databases and screened the titles, abstracts, and full-text articles to select the studies and extract the study data. We included articles published in the English language. The investigators also reviewed the bibliography of the selected articles and previously published systematic reviews. Consensus in cases was achieved if necessary with the assistance of a third author (AA). The following data were extracted from each study: first author, year of publication, study design, number of patients, and baseline characteristics, i.e., sex, age, cirrhosis etiology, and type of surgery.

## 3. Results

### 3.1. A Bit of Embryology

During embryonic life, the digestive system develops from the primitive intestine, which is composed of three parts according to its extent: from the oral cavity to the duodenum, including the liver, bile ducts, and pancreas (anterior primitive intestine); from the terminal part of the duodenum to the proximal 2/3 of the transverse colon (middle primitive intestine); and from the distal 1/3 of the transverse colon to the anal canal through division of the cloacal region (posterior primitive intestine). The primitive midgut elongates rapidly during the first 6 weeks of development to give rise to the primitive bowel loop, which continues to develop partially into the umbilical cord and forms a physiological hernia. Two successive rotations occur at the level of the primitive intestinal loop between the 8th and 10th weeks of development in an anti-clockwise direction (270°) around an axis formed by the superior mesenteric artery (SMA): a first rotation of 90° followed by a second rotation of 180°, allowing the reorganization of the digestive tractus and their integration into the abdominal cavity. After this rotation, the mesos of the transverse colon and the descending colon partially fuse together. The anterior sheet of the transverse mesocolon fuses with the posterior wall of the greater omentum, and the posterior sheet of the descending mesocolon fuses with the retroperitoneal wall [13]. The left colonic angle (LCA) then forms and attaches to the lower pole of the spleen and is located at the border between the structures originating from the primitive midgut and those originating from the primitive hindgut (as shown in Figure 1). The SMA vascularizes the primitive midgut, while the inferior mesenteric artery (IMA) vascularizes the primitive hindgut, respectively. The respect of these embryological planes is essential in the surgical technique of the total removal of the mesocolon for colon cancer [14].

### 3.2. Location and Anatomical Relationships of the Left Colonic Angle (LCA)

The LCA is located deep and high in the left hypochondrium at the level of T11. It forms an acute angle of 40–60° and is totally fixed. Its meso is abutted against the posterior parietal peritoneum, forming the left Toldt fascia. The left phrenicocolic ligament unites it to the diaphragm. It is connected posteriorly with the tail of the pancreas and the left kidney, anteriorly with the greater curvature of the stomach, and superiorly with the spleen and the diaphragm (as shown in Figure 2). Its anatomical position makes its dissection difficult during its release, increasing the risk of mesocolic wounding and devascularization. In addition, the risk of splenic injury that may lead to hemostatic splenectomy is around five-to-six times more frequent compared to other colectomies [15].

### 3.3. Arterial Vascularization of the Left Colonic Angle

#### 3.3.1. What We Know

Given its anatomical location between the transverse and descending colons, arterial vascularization of the left colonic angle is essentially provided by the IMA, including the left colic artery (LCA), and by the SMA via the middle colic artery or artery colica media (ACM) (as shown in Figure 3). The artery colica media is defined as an artery arising from the AMS that vascularizes the transverse colon, and divides into its right and left branches, respectively. Its left branch is intended for vascularization of the left colonic angle. The right and left branches typically arise from a common trunk in 90.9% of cases and separately in the remaining 9.1% of cases, respectively [16].

Concerning the ACSG, several classifications have been described according to its mode of birth on the IMA along with its place and relationship with the inferior mesenteric vein (IMV). Latarjet’s classification [17] separates type I (71% of cases), in which the ASCG and the sigmoid trunk arise separately from the IMV from type II (29% of cases), in which the ASCG and the sigmoid trunk arise from a common trunk of the IMV (Figure 4) [1,18]. Patroni et al. [19] considered the distance between the birth of the ASCG and the IMV at the inferior border of the pancreas (Figure 5) and distinguished subgroup N (66% of cases) if the distance between the two vascular elements was >2 cm and subgroup F (34% of cases) if the distance was <2 cm, respectively. Finally, Miyake et al. individualized four types according to the position of the ACSG in its course to the left colonic angle in relation to the IMV. In type A, the ACGS travels medially to the IMV; in type B, the ASCG is just lateral; in type C, the ASCG is frankly lateral to the IMV; and in type D, the ASCG is absent (as shown in Figure 6) [18].

According to the anatomical work of Griffiths et al. [8], anatomical variations in the arterial vascularization of the left colonic angle are observed in more than one in four patients (28%) due to the border territory between the branches of the SMA and those of the IMA. In 6% of cases, the MCA does not exist, and arterial vascularization is ensured via the MCA; meanwhile, in 22% of cases, the MCA is present, and vascularization of the left colonic angle is ensured by both the MCA and by a branch of the left colic artery. When the CIG (type D) is absent, an accessory middle colonic artery has been described in nearly 80% of cases [20].

#### 3.3.2. What Is Less Known: A New Artery: The Accessory Middle Colonic Artery (AMCA)

To date, there is no consensus on the definition of the accessory middle colonic artery. In the literature, it is often defined as an artery arising from the SMA more proximally than the MCA and runs along the inferior border of the pancreas to vascularize the left colonic angle (Figure 7). In some cases it arises directly from the celiac trunk or even from the hepatic artery [21]. According to this recent study, the AMCA is present from 6.6% to 36.4% of cases, respectively [1,18,22,23]. This highly reported prevalence has been associated with a better knowledge of the vascular anatomy of the left colonic angle due to the advances of preoperative imaging in digestive oncology [12]. Indeed, three-dimensional CT angiography with colonography is an advanced technique that visualizes both the vascularization and the morphology of the colon. It allows an accurate preoperative diagnosis of the location of the colonic tumor and its vascularization, including of its arterial and venous anatomical variants [24]. The advantage of 3D scans is that they enable the tumor, lymph nodes, blood vessels and anatomical relationships to be visualized in all 3 spatial dimensions [11].In addition, several authors have recommended using an additional 2D axial view (thin slice of 0.5–1 mm) to individualize the AMCA as its identification may be difficult when it is small in caliber or fed through the SMA. Interestingly, the prevalence of AMCA varies between European and Japanese subjects (8% versus 48%, respectively), which may suggest, in addition to individual variability, that there is interethnic vascular variability [25,26,27]. Additionally, in the absence of the MCAA, GCA is present in 84.4% to 85.3% of cases [18,20,22]. These data confirm the complementarity of the ACSG and the ACAM in the vascularization of the left colonic angle [18,20,22].

Considering these anatomical variations, Tanaka et al. [16] recently proposed a classification of the arterial vascularization of the left colonic angle into four types based on preoperative imaging. In type 1, vascularization is provided by the left branch of the ACM originating from the common trunk and the ACSG (*n* = 48; 54.5%); in type 2, vascularization is provided by the left branch of the MCA of an independent origin and the ACSG; in type 3, vascularization is provided by the AMCA and the ACSG; and in type 4, vascularization is provided by the ACSG alone. According to this study, which included only 88 patients, the prevalence was 54.5% for type 1 (54.5%), 30.7% for type 3 (30.7%), 9.1% for type 2, and 5.7% for type 1, respectively (Figure 8) [16]. In a recent study, Iguchi et al. [1], who included 96 patients with left colonic angle cancer among 1256 colon cancer patients (7.6%), described five types of arterial vascularization of the left colonic angle based on 3D CT imaging. In type 1A (49%), there was a GCA and a left branch of the MCA from a common trunk observed; in type 2A (27.1%), there was a GCA, a left branch of the MCA from a common trunk, and an MCA from the SMA; in type 3A (16.7%), there was an ACSG and a left MCA arising directly from the AMS; in type 4A (3.1%), there was an ACSG, a left MCA arising directly from the SMA, and an AMCA arising from the SMA; and finally, in type 5A (4.1%), there was an ACSG and an AMCA, and the left MCA was absent(Figure 9). Regardless of these classifications, these studies highlight the variability of arterial anatomy and the non-eligible role that the MCAA plays in the vascularization of the left colonic angle, which could have an impact on the surgical strategies to be implemented for left colonic angle cancers [22].

### 3.4. Venous Vascularization of the Left Colonic Angle

In contrast to studies on arterial vascularization, there is little data available regarding the specific venous drainage of the left colonic angle. Using preoperative 3D CT in 66 colorectal cancer patients, Arimoto et al. [28] characterized the anatomy of the IMV. In this study, the IMV drained directly into the splenic vein in 48.5% of cases, into the MSV in 40.9% of cases, and at the confluence of the two veins in 10.6% of cases, respectively. Additionally, a left colonic angle vein was identified that emptied into the MIV in 93.9% of cases, into the splenic vein in 3.0% of cases, and into the middle colonic vein in a further 3.0% of cases (Figure 2). This junction in the IMV was located below the inferior border of the pancreas in 90.6% of cases and above in 6.3% of cases, respectively. Considerations of the confluence pattern of the IMV could be useful for the safe mobilization of the left colonic angle; the IMV serves as an oncologic landmark in left colonic angle cancer. In Iguchi’s study [1], which evaluated the preoperative vascular anatomy in 96 patients with left colonic angle cancer, a left colonic angle vein was identified in 75% of these patients draining into either the splenic vein or the IMV. The authors therefore proposed a classification consisting of four types (Figure 10). In type 1 (52.1% of cases), the left colonic angle vein drains into the IMV, which then joins the splenic vein; in type 2 (19.8% of cases), the left colonic angle vein drains into the IMV, which then joins the SMV; in type 3 (3.1% of cases), the left colonic angle vein drains directly into the splenic vein; and finally, in type 4 (25%), the left colonic angle vein is absent (Figure 11) [1]. In a recent study by Murono et al. [20], the anatomy of the MCAA and the left colonic angle vein was assessed in 205 patients with colorectal cancer using AngioScan(General Electric Medical Systems, Milwaukee, Wisconsin, USA). A proper left colonic angle vein (LCVA) draining into the IMV was identified in 86.3% of cases, tracing in a total of 95.5% of cases to the inferior border of the pancreas. In this study, the ACSG was the main artery accompanying this vein in 42.4% of cases, followed by the AMCA in 32.7% of cases; no arteries accompanied the vein in 7.8% of cases, respectively. These results suggest that in cases of carcinologic resection of the left colonic angle for cancer, the IMV should be dissected and then tied above the left colonic angle vein outlet, allowing for the resection of the entire length of the corresponding AMCA [20]. Whether central ligation of the MCAA and IMV is even necessary if these vessels are concealed on the dorsal aspect of the pancreas has not been completely elucidated [20]. Indeed, the origin of the MCAA as well as the termination of the IMV and the left colonic angle vein were located behind the pancreas in 20.2%, 31.7%, and 3.9% of cases, respectively.

### 3.5. Lymphatic Drainage of the Left Colonic Angle

The data concerning lymphatic drainage of the left colonic angle are controversial to date. In left colonic angle cancer, lymphatic drainage may occur along the colica media artery, the ACSG, and the AMCA. However, aberrant drainage routes have also been reported in the splenic hilum and pancreas. Given these heterogeneous data, the extension of lymph node adenectomy in left colonic angle cancers remains controversial to date. Moreover, few studies are available due to the relative rarity of these left colonic angle cancers [29].

In other specialties, sentinel node evaluation using radioactive tracers (usually technetium-99 m) has allowed for routine accurate assessments of lymphatic drainage patterns, thereby helping to define the appropriate surgical and therapeutic options [30]. In colorectal surgery, the feasibility of lymphatic mapping has been established, but its role in clinical practice is still being evaluated [31]. Currently, laparoscopic gamma probes are available, allowing for the rapid and minimally invasive assessment of gastrointestinal lymphatic anatomy [32]. A study recently applied this mapping method to investigate the lymphatic drainage pathways of the left colonic angle in vivo in 30 consecutive cancer-free patients. In this study, left colonic angle lymphatic drainage preferentially followed the CIG pathway in the vast majority of cases (>95% at 60 min after injection). However, these results should be interpreted with caution because of the fact that arterial anatomical variants of the left colonic angle were not investigated in this study, such as the presence of an AMCA [31]. In addition, in left colonic angle cancer, lymphatic drainage may be altered due to heavy lymph node infiltration in advanced stages of the disease.

The main data come from the work of Watanabe et al., who evaluated the characteristics of lymphatic flow in left colonic angle cancers using real-time indocyanine green, fluorescent imaging with laparoscopy in 31 patients [29]. Lymphatic flow was visualized in all cases. When an AMCA was identified (which accounted for twelve patients, representing 38.7% of cases), lymphatic drainage was preferentially in three directions: along the AMCA and the ACSG (five patients), along the AMCA (four patients), and along the ACAM and the left branch of the MCA (three patients). In the absence of AMCA, lymphatic drainage occurred along the ACSG in eight patients, along the left branch of the MCA in six patients, and to the root of the IMV in five patients, respectively. In no case was lymphatic drainage shared between the ACSG and the MCA. Finally, regardless of whether there was an AMCA, lymphatic drainage was directed to the origin of the IMV in more than 60% of cases. In the six cases of cancer of the distal third of the transverse colon, lymphatic drainage followed the left branch of the AMCA, while in the seven cases of cancer of the proximal third of the descending colon, lymphatic flow occurred along the ACSG. In this study, no lymphatic flow was directed to the pancreatic tail nor the splenic hilum. Finally, all identified lymph node metastases were located in areas of fluorescently determined lymph flow [29].

In summary, the lymph flow pattern analysis of left colonic angle cancer occurs along three arteries: (i) the left branch of the MCA; (ii) the ACSG; and (iii) along the AMCA when present. Through the study of Watanabe et al., another lymphatic drainage route was identified from the left colonic angle to the root of the IMV in almost 2/3 of their cases.

## 4. Perspectives and Conclusions

In the era of mesocolon resection as the recommended treatment for colon cancer [33], it seems essential to understand the surgical anatomy and its variations. Indeed, this concept allows, as in rectal cancer, the monobloc excision of the colon and its meso, resection at the origin of the feeder vessels, and an adequate lymph node adenectomy. A major pitfall in the application of these principles in the treatment of left colonic angle cancer is the complex regional anatomy, characterized by a very heterogeneous arterial supply. With advances in three-dimensional imaging, a better preoperative understanding of the vascular anatomy of the left colonic angle is now available. If by convention, the colonic angle is mainly vascularized by two arteries (namely the ACGS and the MCA), a third non-negligible source of vascular supply has been identified (the AMCA) in nearly one out of every four patients [22]. Moreover, its prevalence is multiplied by 3.6 in the absence of an ACSG, underlining the complementarity of these vessels. Finally, the AMCA has also been identified as a lymphatic drainage pathway for cancers of the left colonic angle [29].

If the arterial anatomy of the left colonic angle has been classified into five types and the venous anatomy into four types, preoperative identification of this complex anatomy could allow for an adequate, minimally invasive oncologic resection with minimal intraoperative bleeding complications [34]. However, not only the number of studies, but also the number of patients included is limited. Moreover, the official terminology and definition of the MCAA has not been validated as of yet [1]. These anatomical variations of the left colonic angle plead for carcinologic surgical resection adapted to each patient, and not standardized [5,6,7] based on preoperative (Figure 12) and/or intraoperative identification.

## Figures and Tables

**Figure 1 jpm-13-01198-f001:**
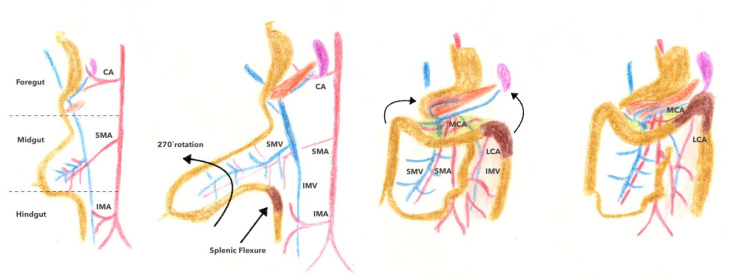
Embryological development of the primitive intestinal tract. CA: celiac artery; SMA: superior mesenteric artery; IMA: inferior mesenteric artery; MCA: middle colic artery; LCA: left colic artery; SMV: superior mesenteric vein; IMV: inferior mesenteric vein.

**Figure 2 jpm-13-01198-f002:**
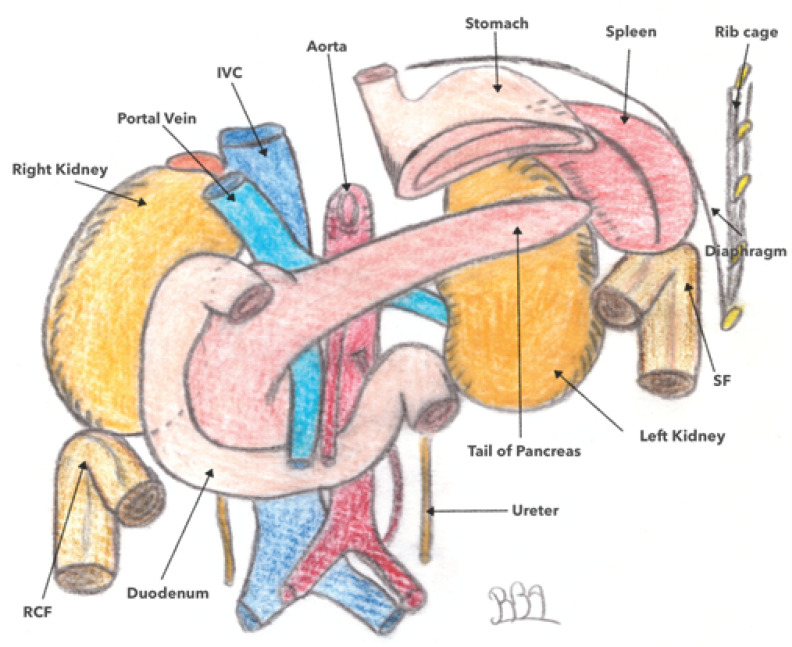
Anatomical siting of splenic flexure. SF: splenic flexure; RCF: right colic flexure; IVC: inferior vena cava.

**Figure 3 jpm-13-01198-f003:**
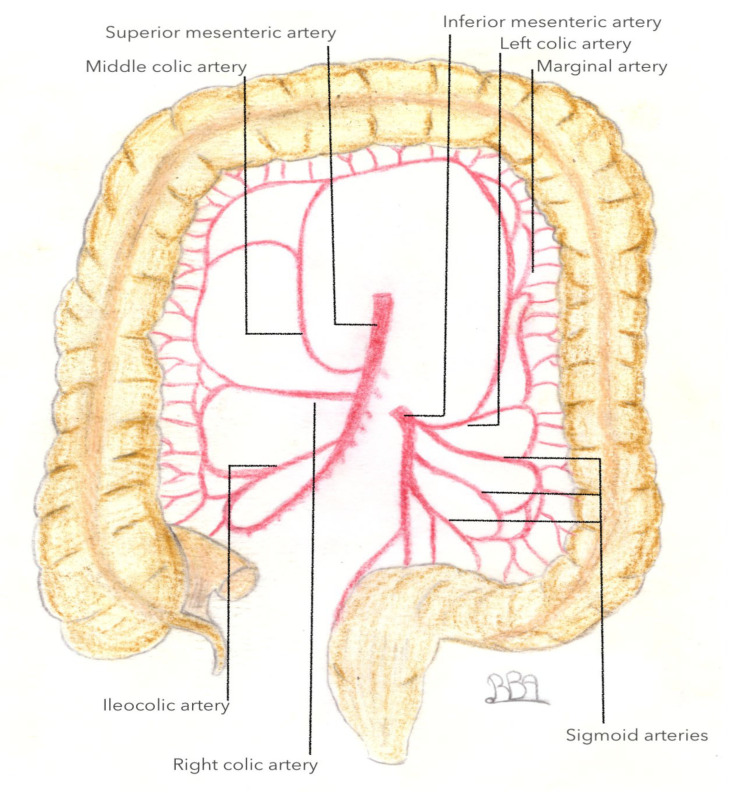
Blood supply of large intestine.

**Figure 4 jpm-13-01198-f004:**
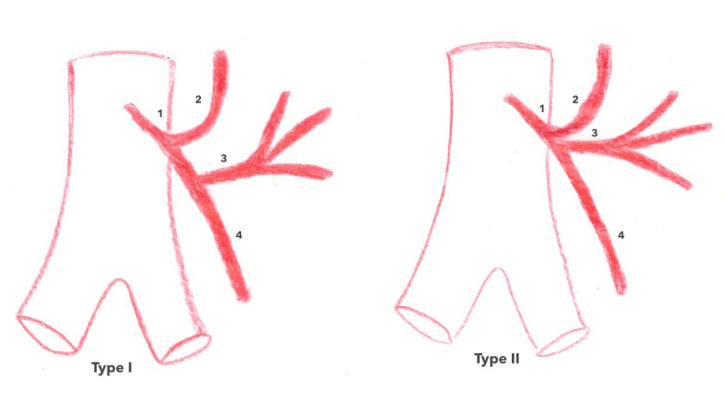
Latarjet’s classification of the IMA branching pattern. IMA: inferior mesenteric artery.

**Figure 5 jpm-13-01198-f005:**
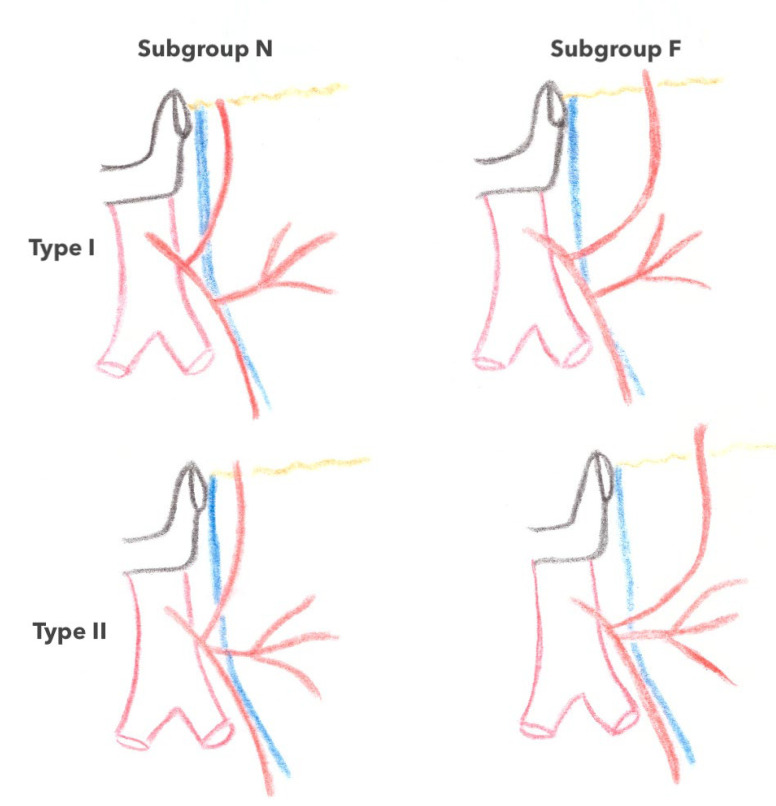
Patroni et al. [19] AI classification based on the positional relationship between the LCA and IMV. Types I and II represent spread-out or fan-shaped IMA (inferior mesenteric artery) branching patterns, respectively. Subgroups F and N represent an IMV-LCA (inferior mesenteric vein-left colic artery) distance of more or less than 20mm, as the inferior border of the pancreas.

**Figure 6 jpm-13-01198-f006:**
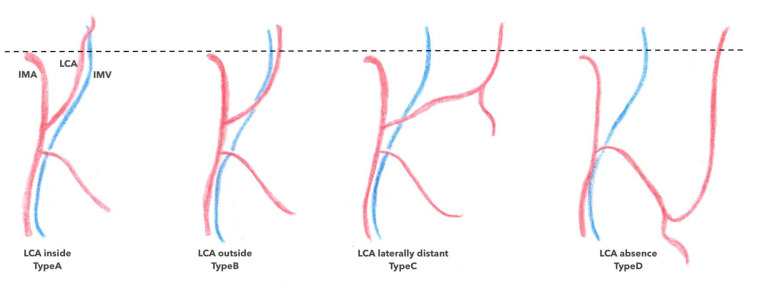
Miyake and al classification 2018. LCA (left colic artery) type A; LCA outside Type B; LCA laterally distant Type C; LCA absence Type D.

**Figure 7 jpm-13-01198-f007:**
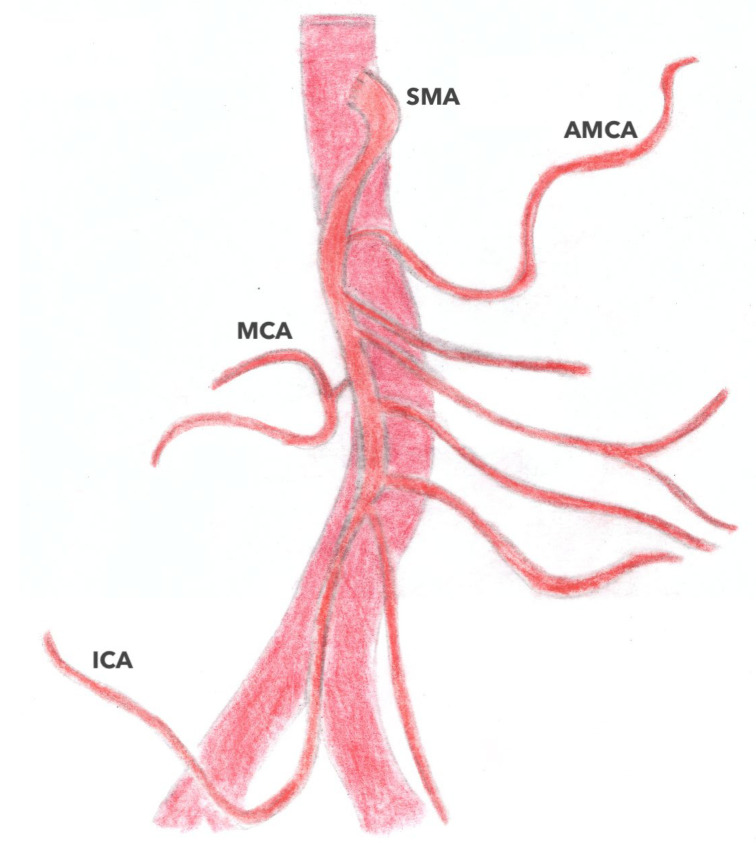
Vascular branching pattern of the AMCA. AMCA accessory middle colic artery; ICA: ileocolic artery; MCA: middle colic artery; RCA right colic artery; SMA: superior mesenteric artery.

**Figure 8 jpm-13-01198-f008:**
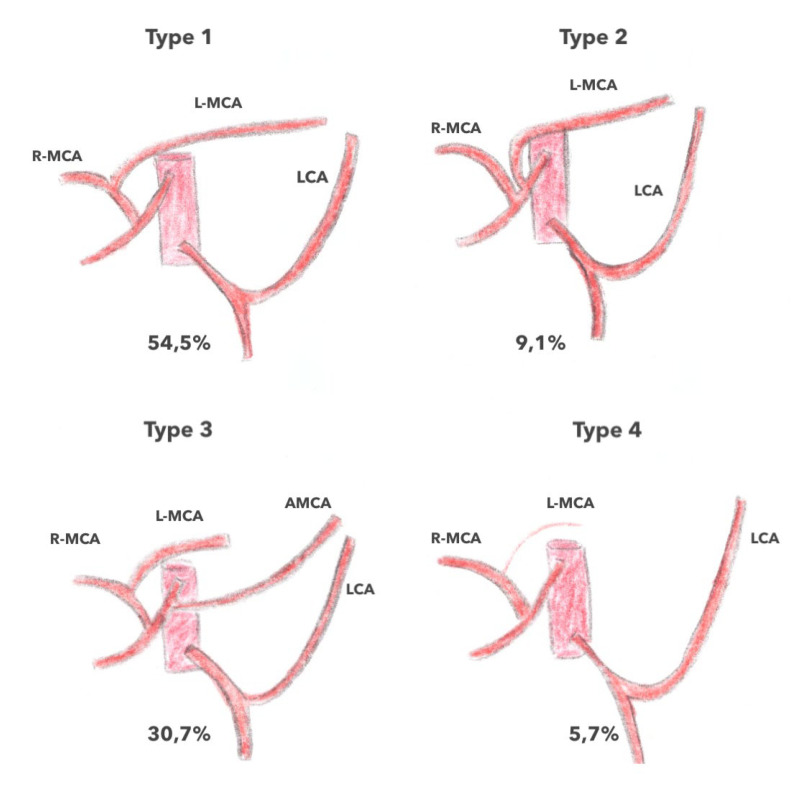
The branching pattern of the blood supply to the splenic flexure according to Tanka et al., 2019. Type 1, the left branch of the MCA (middle colic artery) and the LCA (left colic artery) 54.5%; Type 2, the left branch of the MCA, arising directly from the SMA and the LCA 9.1%; Type 3, the A-MCA and the LCA 30.7%; Type 4, the LCA alone 5.7%.

**Figure 9 jpm-13-01198-f009:**
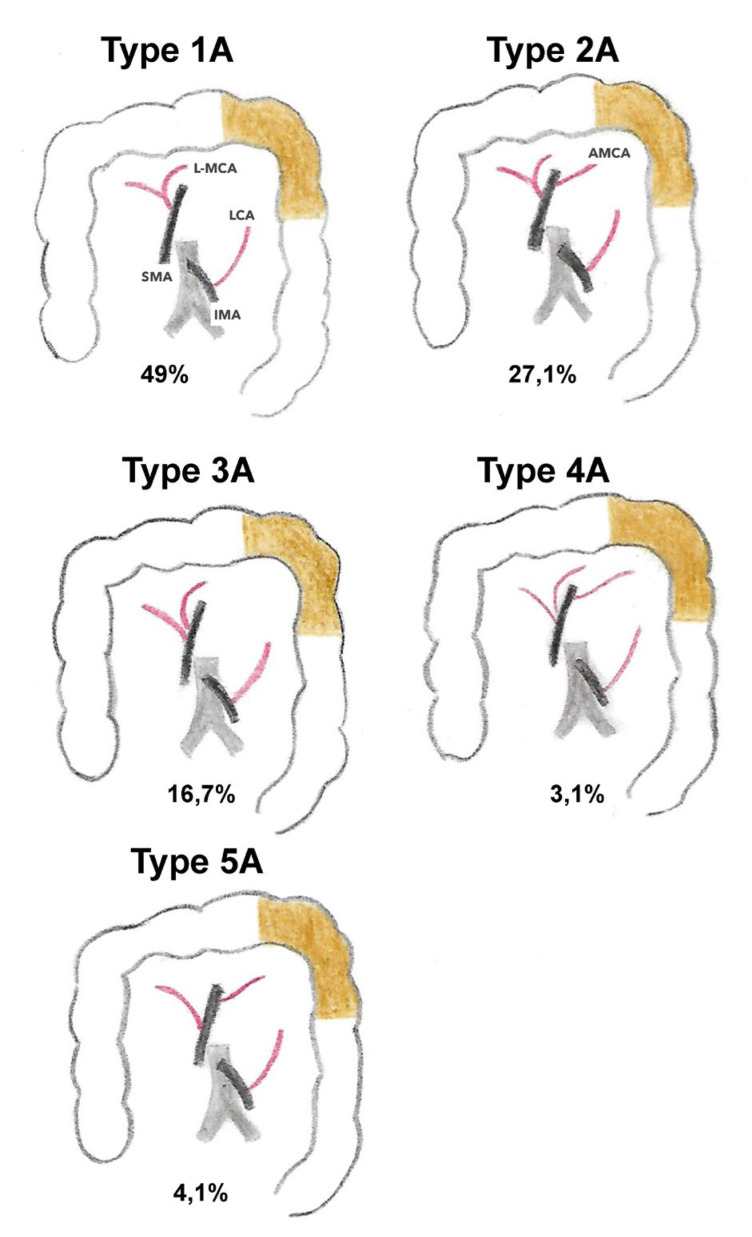
The branching pattern of blood supply to the splenic flexure according to Iguchi et al., 2020. SMA: superior mesenteric artery; IMA: inferior mesenteric artery; L-MCA: left branch middle colic artery; LCA: left colic artery; AMCA: accessory middle colic artery. Type 1A: 49%; Type 2A: 27.1%; Type 3A: 16.7%; Type 4A: 3.1%; Type 5A: 4.1%.

**Figure 10 jpm-13-01198-f010:**
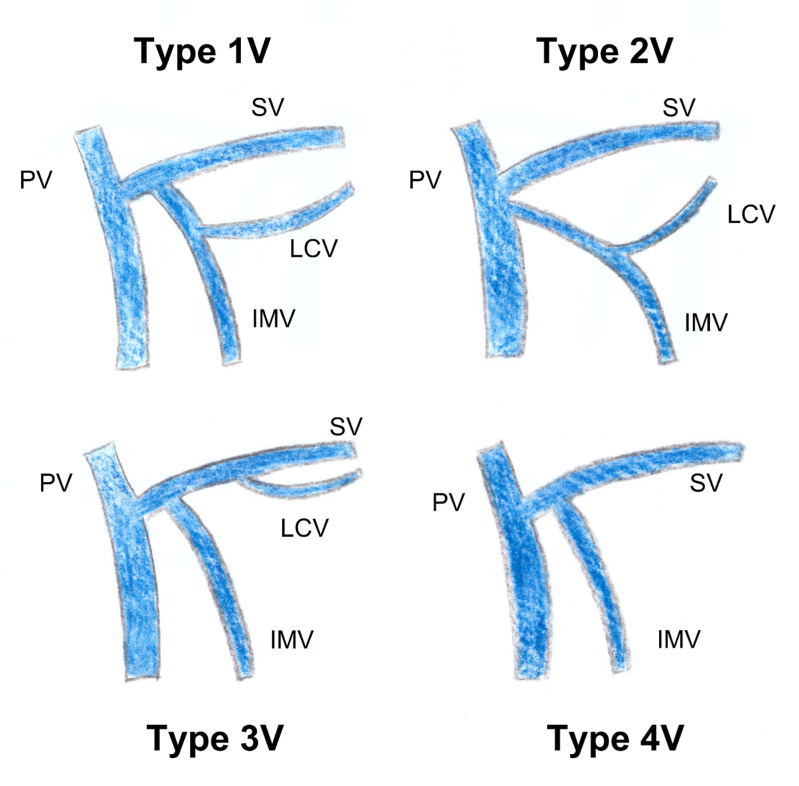
Vein branching patterns of the LCV according to Iguchi et al. 2020. PV: portal vein; SV: splenic vein; LCV: left colic vein; IMV inferior mesenteric vein.

**Figure 11 jpm-13-01198-f011:**
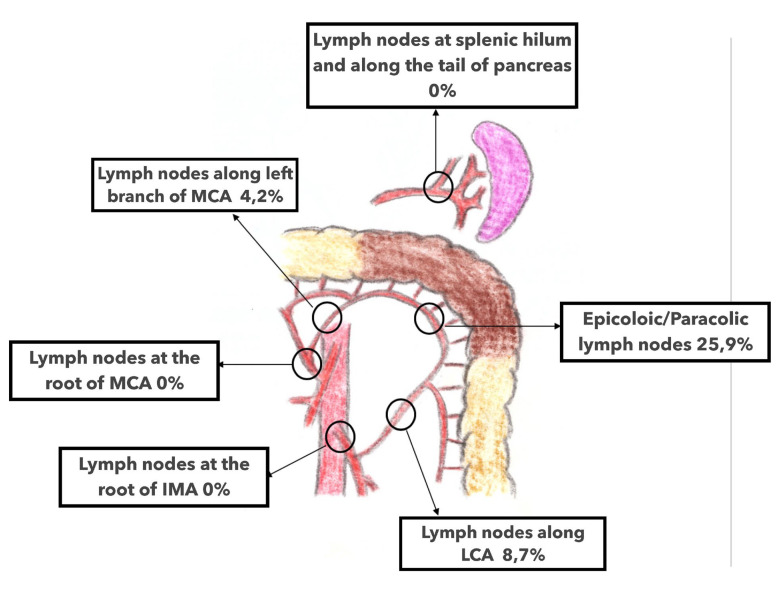
Distribution of lymph node metastasis according to Nakagoe and al. 2021. Lymph nodes at splenic hilum and along the tail of pancreas 0%; epicolic/paracolic lymph nodes 25.9%; lymph nodes along LCA (left colic artery) 8.7%; lymph nodes at the root of IMA (inferior mesenteric artery) 0%; lymph nodes along left branch of MCA (middle colic artery) 4.2%.

**Figure 12 jpm-13-01198-f012:**
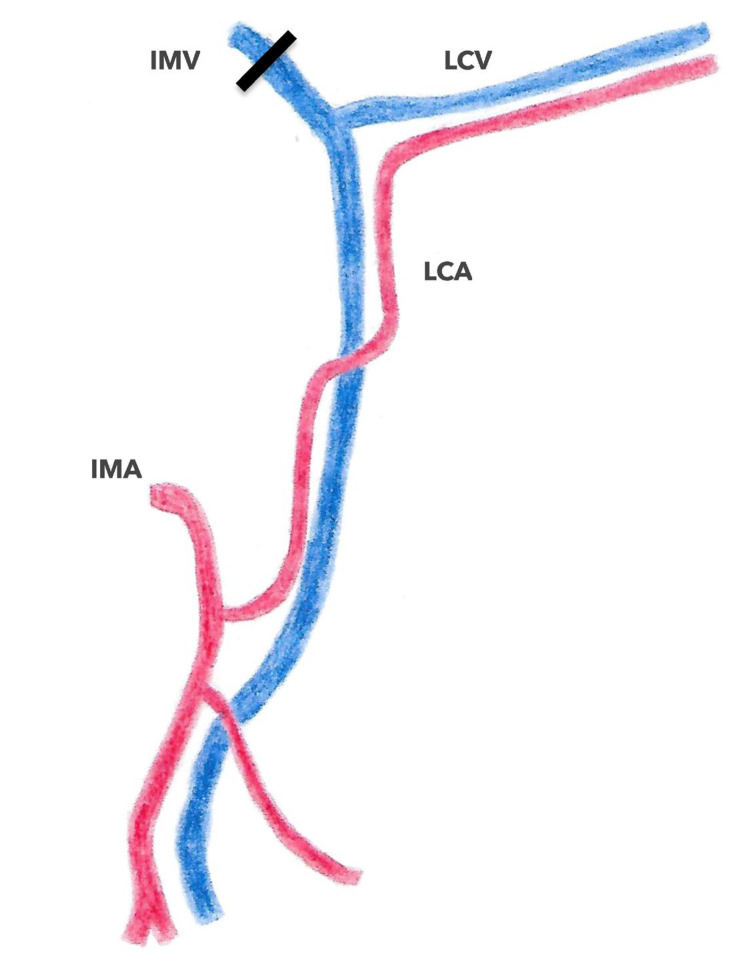
Dissection level of IMV (inferior mesenteric vein) for splenic flexure cancer according to Murono and al. 2019. IMV (inferior mesenteric vein; LCV: left colic vein; IMA: inferior mesenteric artery; LCA: left colic artery.

## Data Availability

Not applicable.

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
