# Peer review of "Revisited Surgical Anatomy of the Left Colonic Angle for Tailored Carcinologic Colectomy: A Review"

_jpm, 2023, doi:10.3390/jpm13081198_

Round 1

Reviewer 1 Report

The paper is a review about the surgical anatomy for splenic flexure tumors. The paper is clearly presented, the bibliography is adequate and recent, and most of the studies are cited. There are some comments that I wish the authors could address: 

- This is a nice literature review, but a systematic review could be a better option.

- The aim of the paper is not clear. In the abstract, in the methods section, it is stated: "We performed a methodological review of studying anatomical variations of left colonic artery". However, in the title and introduction, the aim is the anatomy of the left colonic angle. 

- In the abstract, the methods section is too short and does not describe the paper's questions.

- All the images are annotated in French, while the paper is in English. 

- The terminology used is not familiar in English literature. For example, left superior colic artery is called Left colic artery in English literature. 

- Acronyms are not homogeneous in the article. Superior mesenteric artery is sometimes MSA, and SMA. The same is almost for all described vessels. 

- There are some mistakes in the description of classifications. For example in page 8: “In 6% of the cases, the MCA does not exist, and arterial vascularization is ensured via the MCA, whereas in 22% of the cases, the MCA is absent, and vascularization of the left colonic angle is ensured both by the MCA and by a branch of the right superior colonic artery.”

- The same in page 9: "Thus, in the absence of MCAAs, GCA is present in 84.4% of cases; conversely, GCA can reach 85.3% of cases in the absence of GCA". 

- There are some sentences that should not be there. Example in page 10:  "(Figure 9 to be done by Roukaya)" or in page 12: "(Figure 12 to be taken Figure 3 from Mur0no)"

- The authors highlighted the importance of modern CT-scans to identify the vascular anatomy. It would be interesting to have the variations of vascular anatomy shown in CT-scan images or 3D reconstructions from CT-scans.

- In the abstract, in the methods section it is stated: "We performed a methodological review of studying anatomical variations of left colonic artery". However, in the title and introduction, the aim is the anatomy of the left colonic angle. 

Author Response

JPM, manuscript jpm-2482147

Reviewers' comments:

Reviewer(s)' Comments to Author:

Reviewer #1 (RW1):

The paper is a review about the surgical anatomy for splenic flexure tumors. The paper is clearly presented, the bibliography is adequate and recent, and most of the studies are cited. There are some comments that I wish the authors could address: 

RW1.1- This is a nice literature review, but a systematic review could be a better option.

Authors’ response:  Thank you for this comment which contributed to upgrade our academic work. However, literature is scarce about this topic and that is why we only performed review and not systematic review.

RW1.2- The aim of the paper is not clear. In the abstract, in the methods section, it is stated: "We performed a methodological review of studying anatomical variations of left colonic artery". However, in the title and introduction, the aim is the anatomy of the left colonic angle. 

Authors’ response:  Thank you for this comment which contributed to upgrade our academic work.

Actions in the manuscript: We change the sentence in the title and introduction

RW1.3- In the abstract, the methods section is too short and does not describe the paper's questions.

Authors’ response:  Thank you for this comment which contributed to upgrade our academic work.

Actions in the manuscript: we change the method section in the manuscript.

RW1.4- All the images are annotated in French, while the paper is in English. 

Authors’ response:  Thank you for this comment which contributed to upgrade our academic work.

Actions in the manuscript: we changed the annotation in all images

RW1.5- The terminology used is not familiar in English literature. For example, left superior colic artery is called Left colic artery in English literature. 

Authors’ response:  Thank you for this comment which contributed to upgrade our academic work.

Actions in the manuscript: we change the terminology in the manuscript for left superior colic artery into left colic artery

RW1.6- Acronyms are not homogeneous in the article. Superior mesenteric artery is sometimes MSA, and SMA. The same is almost for all described vessels. 

Authors’ response:  Thank you for this comment which contributed to upgrade improve our academic work.

Actions in the manuscript: we changed the acronyms in the manuscript

RW1.7- There are some mistakes in the description of classifications. For example in page 8: “In 6% of the cases, the MCA does not exist, and arterial vascularization is ensured via the MCA, whereas in 22% of the cases, the MCA is absent, and vascularization of the left colonic angle is ensured both by the MCA and by a branch of the right superior colonic artery.”

Authors’ response:  Thank you for this comment which contributed to upgrade our academic work.

Actions in the manuscript: We changed the sentence in the manuscript

RW1.8The same in page 9: "Thus, in the absence of MCAAs, GCA is present in 84.4% of cases; conversely, GCA can reach 85.3% of cases in the absence of GCA". 

Authors’ response:  Thank you for this comment which contributed to upgrade improve our academic work.

Actions in the manuscript: We changed the sentence in the manuscript

RW1.9- There are some sentences that should not be there. Example in page 10:  "(Figure 9 to be done by Roukaya)" or in page 12: "(Figure 12 to be taken Figure 3 from Mur0no)"

Authors’ response:  Thank you for this comment which contributed to upgrade improve our academic work.

Actions in the manuscript: We deleted the sentences in the manuscript

RW1.10- In the abstract, in the methods section it is stated: "We performed a methodological review of studying anatomical variations of left colonic artery". However, in the title and introduction, the aim is the anatomy of the left colonic angle. 

Authors’ response:  Thank you for this comment which contributed to upgrade improve our academic work.

Actions in the manuscript: We changed the sentence in the manuscript.

Reviewer 2 Report

1.       Figure legends are only visible in some figures; providing a clear figure legend makes manuscripts look better.

2.        Figure-6 legend, check the reference style.

3.       Provide the methodology after the introduction in the manuscript.

4.       What is the incident rate of left colonic angles in colon cancer? Provide the table about the incidence of left colonic angles in colon cancer.

5.       Check the figures in the manuscript; they need to be more straightforward and use some software to draw the images that make a better version of the manuscript.

Author Response

Reviewer #2 (RW2):

RW2.1 Figure legends are only visible in some figures; providing a clear figure legend makes manuscripts look better.

Authors’ response: Thank you for this comment which contributed to upgrade improve the manuscript.

Actions in the manuscript: figures were checked

RW2.2 Figure-6 legend, check the reference style.

Authors’ response: Thank you for this comment which contributed to upgrade improve the manuscript.

Actions in the manuscript: reference was checked

RW2.3 Provide the methodology after the introduction in the manuscript.

Authors’ response: Thank you for this comment which contributed to upgrade improve the manuscript.

Actions in the manuscript:

RW2.4 What is the incident rate of left colonic angles in colon cancer? Provide the table about the incidence of left colonic angles in colon cancer.

Authors’ response: Thank you for this comment which contributed to upgrade improve the manuscript.

Actions in the manuscript: tableau des pros du colon (Prs A/S/M)

RW2.5 Check the figures in the manuscript; they need to be more straightforward and use some software to draw the images that make a better version of the manuscript.

Authors’ response: Thank you for this comment which contributed to upgrade improve the manuscript.

Actions in the manuscript: As suggested  by the reviewer, figure has been upgraded
